# Cross-modal fusion of brain imaging and clinical data for Parkinson's disease progression prediction

Jinyu Wen[1], Amei Chen[2], Jingxin Liu[3], Hua Xiong[4], Meie Fang[1]*, Xinhua Wei[2]

1 School of Computer Science and Cyber Engineering, Guangzhou University, Guangzhou, Guangdong, China, 2 Department of Radiology, Second Affiliated Hospital of South China University of Technology, Guangzhou, Guangdong, China, 3 School of Mathematics and Physics Sciences, Chongqing University of Science and Technology, Chongqing, Chongqing, China, 4 Department of Radiology, Shapingba District People's Hospital of Chongqing, Chongqing, Chongqing, China

* fme@gzhu.edu.cn

## Abstract

**Background:** Machine learning shows great potential in science but struggles with complex, high-dimensional multi-omics data. PD progression is long, diagnosed mainly by clinical signs. This paper proposes a novel decision fusion method to improve the precision of the classification of progression of PD using imaging with clinical data.

**Methods:** A Cross-Modal Fusion Prediction Model (CMFP) is proposed, with key steps that involve data preparation, modelling, and prediction. The data encompasses three modalities: clinical, DTI (diffusion tensor imaging), and DAT (dopamine transporter), with Lasso used for the selection of features. Individual modalities are classified using AdaBoost and the results are integrated into the new fusion strategy, CMF, to obtain a novel model. Finally, this model is used for predictions.

**Results:** The predictive performance of CMFP on the progression of PD achieved an AUC of 77.91%. This represents improvements of 24.48%, 30.78%, and 32.7% in AUC compared to predictions solely with clinical data, DTI data and DAT data, respectively. The combined prediction of clinical and DTI data demonstrated statistical significance compared to predictions based solely on clinical data, with a p-value of 9.183e-4. Additionally, this method identified crucial brain regions and important clinical metrics associated with PD. It should be noted that using the DTI metric along the perivascular space (DTI-ALPS) to predict and evaluate the progression of PD has relatively more advantages compared to DTI-clinical fusion prediction. Among them, the ACC can increase by 3.85%.

**Conclusion:** The results indicate that CMFP is effective, contributing to overcoming the limitations of low predictive performance in single-modal data and enhancing the accuracy of the PD progression predictions.

**Data availability statement:** The format has been improved according to the requirements of the journal and the code has also been uploaded to Dryad Digital Repository (DOI: 10.5061/dryad.63xsj3vdv), which can be found at http://datadryad.org/share/zvyCwl-SD-ApTC7IwAXSLJJxA1W53rjXR_c1xPykaxM.

**Funding:** This work was supported in part by the National Natural Science Foundation of China (No. 62072126, No. 62506086), in part by the Fundamental Research Projects Jointly Funded by Guangzhou Council and Municipal Universities No. SL2023A03J00639, in part by the Key Laboratory of Philosophy and Social Sciences in Guangdong Province of Maritime Silk Road of Guangzhou University (GD22TWCXGC15), in part by the Natural Science Foundation of Chongqing (No. CSTB2024NSCQ-MSX1087), and the Guangxi Science and Technology Program (No. AD23023001).

**Competing interests:** The authors declare no conflict of interest.

## Introduction

Parkinson's Disease (PD) is a prevalent progressive neurodegenerative disorder characterised by tremors at rest, bradykinesia, rigidity, and postural instability, predominantly affecting the human nervous system and motor control [1]. The prolonged progression of the disease significantly affects quality of life, leading to physical disabilities and nonmotor symptoms, and is associated with increased mortality rates. In such scenarios, early prediction of the disease is crucial for implementing appropriate intervention measures [2]. However, there is currently a lack of reliable clinical outcomes and/or biomarkers for progression, with clinical evaluations potentially being time-consuming and subject to variations based on patient conditions [3]. Although conventional MRI excels at providing tissue contrast, its value for suspected patients with PD mainly lies in excluding concurrent brain diseases rather than confirming PD diagnoses [4]. Given the heterogeneity of PD, which can be categorised into various subtypes based on the age of onset, clinical manifestations, and progression speed [5], predicting the progression of PD becomes especially imperative.

Several studies have indicated that changes in white matter microstructure occur before the loss of cortical neurones, even in the absence of apparent grey matter atrophy. DTI technology has been widely used to assess microstructural damage in white matter in patients with PD [6–8]. For example, studies have suggested that the measurement of the free water content in the substantia nigra using DTI can predict changes in motor slowness and cognitive status in PD patients in the next year [9]. A recent study using DTI revealed significant changes in the results of the whole brain voxel of fractional anisotropy (FA) and mean diffusivity (MD) of the corpus callosum, especially in its regions of the knee and body. Moreover, the decrease in corpus callosum FA was closely related to the decline in FA and MD in widespread cortical and subcortical regions [10]. Multivariate regression analysis further confirmed a negative correlation between corpus callosum FA values and the severity of motor stiffness in PD patients, with the strongest impact observed in the anterior part of the corpus callosum [10]. These findings highlight the potential of corpus callosum microstructural changes as biomarkers for motor stiffness symptoms and disease progression, even in the early stages of PD.

Machine learning (ML) stands as a promising pivotal technology in the prediction domain [11]. ML, especially when combined with data mining techniques, is devoted to advancing algorithms that learn patterns from known data to form models, which are then applied to unknown data to forecast outcomes [12,13]. Consequently, ML has been extensively applied to predict PD and its progression, with the aim of improving its performance [5,14–17]. Currently, statistical models that predict the clinical progression of PD present challenges. Previous univariate longitudinal or multivariate analyses from cross-sectional studies have limitations in predicting individual outcomes or specific time points [18,19]. Existing research has shown that the construction of multimodal and hybrid models can facilitate the exploration of progression of PD [20]. In this paper, we used the ML technique and proposed a cross-modal fusion decision-making approach to address the limitations of low predictive performance in single-modal data.

In investigating the mechanisms of progression of PD, Yang and his team focused on changes in brain structural connectivity, revealing significant structural pattern transitions in eight core regions of the cerebral cortex and subcortex as PD progressed [21]. Simultaneously, Jain and his team successfully built an efficient model to predict the Unified Parkinson's Disease Rating Scale (UPDRS) using collected noninvasive language test data, helping to improve remote monitoring of the progression of PD [22]. Although DTI data in the entire brain have significant potential to describe early microstructural changes in PD, research that integrates DTI with clinical characteristics to predict PD progression is relatively limited. Our work makes the following contributions:

(1) A cross-modal fusion prediction method (CFMP), which was used to predict the progression of Parkinson's Disease. The AUC values of fusion prediction using the CFMP are improved by 24.48% and 30.78% compared to using only clinical or DTI data to predict the progression of PD, respectively.

(2) This paper reveals key brain regions and important clinical characteristics that are closely associated with the pathological progression of Parkinson's Disease. The results of the prediction of fusion using selected key clinical and DTI data show an improvement of 13.82% in AUC compared to the use of all combinations of characteristics for the prediction.

(3) Using the DTI metric along the perivascular space (DTI-ALPS) to predict and evaluate the progression of PD has relatively more advantages compared to DTI-clinical fusion prediction. Among them, the ACC can increase by 3.85%.

## Methods

### Data acquisition

This research relied on the Parkinson's Progression Markers Initiative (PPMI) database, which contains longitudinal data from patients with PD in multiple centres. All PPMI sites have obtained approval from their respective ethics committees, and all PPMI participants provided their written informed consent prior to participation. The severity of PD was assessed using the Hoehn and Yahr Scale (HYS), typically based on factors such as decline in motor skills, reduced quality of life, and dopaminergic losses, categorising PD into five stages (stages 1-5). Inclusion criteria comprised: 1) patients with PD with complete HYS scores followed longitudinally for at least 5 years; 2) baseline data that included comprehensive MRI images (3D T1WI and DTI images); 3) MRI scans performed on a 3T Siemens MRI machine. Exclusion criteria included: 1) incomplete clinical data; 2) poor image quality or errors in image processing.

A total of 123 patients with PD were included in the study. By assessing the longitudinal changes in the HYS scores of these patients over 5 years in the PPMI database, it was found that 74 patients had scores higher than baseline, 46 had scores identical to baseline and 3 had scores lower than baseline. In this research, patients with HYS scores higher than baseline were classified into the progression group (n = 74), while those with scores the same as or lower than baseline were classified into the stable group (n = 49). The research also encompassed extensive clinical data, including gender, age, years of education, PD motor assessment scales, and cerebrospinal fluid markers: A$\beta$42, a-syn, t-tau, and p-tau. The 123 participants underwent non-contrast-enhanced 3D volumetric T1-weighted MRI and DTI scans using a 3T Siemens MRI scanner. For DTI image pre-processing, the PANDA software was employed, which primarily involved several steps: 1. Removal of non-brain tissues; 2. Correction for eddy current effects and minor head motion; 3. Computation of the diffusion tensor matrix based on voxels; 4. Fibre allocation using the FACT algorithm to produce deterministic fibre tractography. If the angle of curvature exceeded 45 ° or the FA of the voxel was less than 0.2, the trajectory was terminated. In the ML algorithm modelling, we emphasised the selection of baseline clinical metrics, DAT markers, and DTI brain network metrics related to the progression of PD as features. Data screening was performed using the Lasso method to improve model accuracy and generalisability.

### Cross-Modal Fusion prediction model: CMFP

We used baseline clinical metrics, DAT markers, and DTI white matter MD (50) to separately train machine learning models to predict the progression of PD five years later. In the process of single-modal modelling, we utilised the AdaBoost

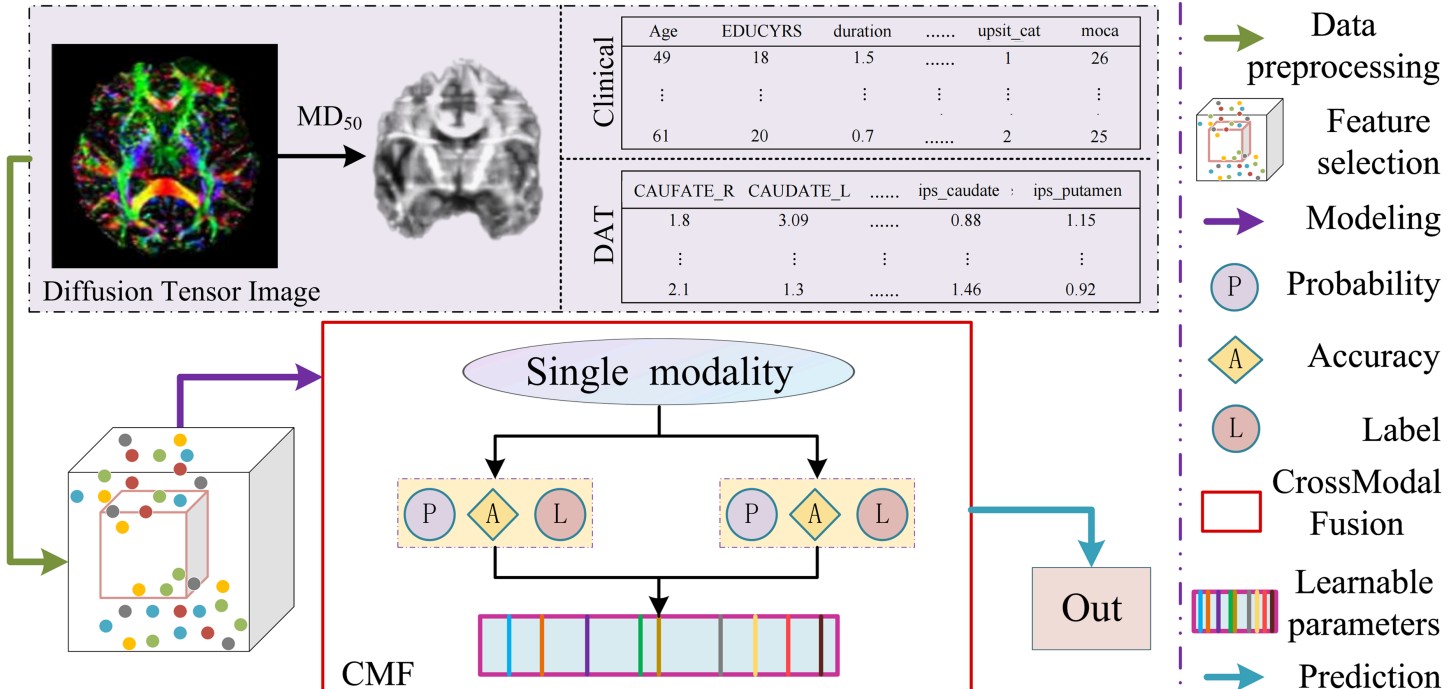

**Fig 1**. **The frame of the method in this paper is composed of four parts: data preprocessing, feature selection, classifier modeling and prediction.**

algorithm (Adaptive Boosting) for classification. AdaBoost is a boost method that combines multiple weak classifiers to form a strong classifier. Its adaptiveness lies in the fact that the weights of the samples misclassified by the previous weak classifier (corresponding to the weights of the samples) are enhanced. After the weights are updated, these samples are used again to train the next weak classifier. In each training round, a new weak classifier is trained on the entire dataset, generating new sample weights and the influence of that weak classifier. This process iterates until a predetermined error rate is achieved or a specified maximum number of iterations is reached. AdaBoost can adaptively adjust the assumed error rate based on feedback from weak classifiers, demonstrating high efficiency. Some researchers have used AdaBoost for early prediction of PD [23].

To further improve the performance of univariate baseline data in predicting progress after 5 years, we adopt a cross-modal data combination approach and propose a new decision strategy to improve predictive performance. Fig 1 is the framework of this paper, including data pre-processing, feature selection, modelling, and prediction. The specific steps of CMFP are as follows:

1. Set the epoch parameter to 120. Adam is used as the optimisation algorithm. The initial learning rate is 0.01, and the learning rate decreases by 0.1 every 30 epoch. The calculation of the loss adopts the computation process of the Mean Squared Error (MSE).
2. The dataset was first randomly partitioned into training sets (60%), validation sets (30%) and test sets (10%). Subsequently, the selection of features using the Lasso method was performed exclusively on the training set, ensuring that no information leakage was induced from the validation or test sets.

3. The training set data of different modes were trained by AdaBoost, and the obtained models were predicted by verification set data. The probability, label and accuracy of prediction were obtained, and the single-modal trained models were saved, respectively.

4. The results of different single-modal predictions (probability, label, and accuracy) were input into the multimodal fusion algorithm (CMF: Cross-Modal Fusion), a set of new prediction results (probability, label and accuracy) were obtained, and the fusion model was saved;

5. Repeat 2-5 steps until epoch=120;

6. Find the model with the highest accuracy saved during training of single-modal and cross-modal fusion, as *Ms* and *Mc*;

7. Input the test set data into model *Ms* to obtain predicted results (probability, predicted label, and accuracy), then input these results into model *Mc* for testing to obtain the fused prediction probability and labels from the two modalities.

Original input clinical data included 16 clinical variables, including 3 basic information (age, gender, and years of education), 9 motor and cognitive scores (UPDRS Part I score, UPDRS Part II score, UPDRS Part III score, Total UPDRS score, ESS score, GDS score, UPSIT score, RBDSQ score, MoCA score), and 4 Cerebrospinal fluid (CSF) protein concentrations ($A\beta42$, a-syn, t-tau, p-tau). We used the Lasso method for data selection. As shown in Table 1, we pick four clinical variables (age, UPDRS Part III Score, UPDRS Total Score, UPSIT score) with high correlation, which represent age, a score to evaluate motor function in PD, total score of the Unified PD Rating Scale (UPDRS) and the University of Pennsylvania Smell Identification Test, respectively. Among them, the UPDRS provides a comprehensive and detailed assessment of the severity of the disease. UPSIT score refers to the University of Pennsylvania Smell Recognition Test. For DTI variables, we used the MD value of the brain region to predict and 7 characteristics (Splenium of Corpus Callosum, Fornix (Column and Body of Fornix), Inferior Cerebellar Peduncle (left), Superior Cerebellar Peduncle (left), Fornix (Crescent)/Straxia Terminalis (right), Tapetum Right, Tapetum Left). They represent seven different regions of the brain. For DAT data, four characteristics (left putamen, high putamen, low striatum, high striatum) were finally selected, which represent the left caudate nucleus, the right caudate nucleus, high putamen, low putamen, high striatum, low striatum, respectively. These nuclei are closely related to the pathogenesis of PD. In addition, in patients with PD, dopamine deficiency in the striatum leads to the corresponding motor symptoms.

In our proposed prediction method, a critical component lies in the decision strategy (CMF) (specific steps as Algorithm 1). By applying CMF to individual predictions derived from single-modal data, we obtain the final prediction result (the predicted labels and their associated probabilities).

Before the decision-making process, we obtain predictions from two single-modal datasets, including predicted labels, predicted probabilities, and accuracy values. We denote them as $\rho_a$, $\rho_b$, $\eta_a$, $\eta_b$, $\varphi_a$ and $\varphi_b$, respectively. The single-modal prediction results are then processed through our proposed decision strategy CMF to generate the final fused prediction result $\Phi$, which comprises both the predicted probability and label, i.e., $\Phi = [\rho, \eta]$.

We define three key metrics for the fusion process: category difference: $\Omega = \rho_a - \rho_b$ (the difference between predicted categories from two modalities); accuracy difference sign: $\sigma = \varphi_a - \varphi_b$ (indicating which modality has higher accuracy);

**Table 1**. The features of the strongly correlated combination were selected from three datasets (Clinical, DTI, and DAT) using the Lasso method.

| Clinical | UPDRS Part III Score | UPDRS Total Score | Age | UPSIT score |
|---|---|---|---|---|
| DTI | Splenium of Corpus Callosum | Fornix (Column and Body of Fornix) | Tapetum Right | |
| | Inferior Cerebellar Peduncle (Left) | Superior Cerebellar Peduncle (Left) | Tapetum Left | |
| | Fornix (Crescent)/Stria Terminalis (Right) | | | |
| DAT | Putamen Left | High Putamen | Low Striatum | High Striatum |

**Algorithm 1 : Cross-Modal Fusion (CMF).**

```
 1: Input:
 2:    Probability vectors: η_a, η_b
 3:    Class vectors: ρ_a, ρ_b
 4:    Accuracy values: φ_a, φ_b
 5: Output:
 6:    Fused probability and class: Φ
 7:
 8: l ← length of η_a
 9: α ← 0.4
10: ι ← 1
11: for i ← 1 to l do
12:    Ω ← ρ_a − ρ_b
13:    σ ← φ_a − φ_b
14:    if Ω = 0 then
15:      if σ > 0 then
16:         η_new ← η_a
17:         ρ_new ← ρ_a
18:      else
19:         η_new ← η_b
```

```
 1:         ρ_new ← ρ_b
 2:      end if
 3:    else
 4:      if σ > 0 then
 5:         υ ← η_a − η_b
 6:         η_new ← ι/(1 + e^{−α∗υ}) + η_a
 7:      else
 8:         υ ← η_b − η_a
 9:         η_new ← ι/(1 + e^{−α∗υ}) + η_b
10:      end if
11:      if η_new < 0.5 then
12:         ρ_new ← 0
13:      else
14:         ρ_new ← 1
15:      end if
16:    end if
17:    Φ ← [ρ_new, η_new]
18: end for
19: return Φ
```

probability difference: $\upsilon = \eta_a - \eta_b$ (the difference between predicted probabilities). The final predicted labels and probabilities are determined through different decision branches based on the values of $\Omega$, $\sigma$, and $\upsilon$. The probability adjustment uses the formula $\eta_{new} = \iota/(1 + e^{-\alpha*\upsilon})$, where, $\alpha$ is set to 0.4 and $\iota$ is set to 1.

For the entire algorithm covered in this paper, we are programming in Python 3.7.9 with Pytorch, running on the Linux platform (all experiments are carried out on the GNU/Linux x86 64 system of GeForce RTX 3090 Ti 12 Intel Core Interl(R) Xeon(R) CPU E5-2678 v3 2.50GHz 64GB RAM device).

## Ethics

For the PPMI data used in this research, all participants provided written informed consent approved by the institutional review board of each participating institution.

## Statistics

To evaluate the effectiveness of the CMFP method, this study compared the performance of the model with single or dual feature prediction models and different prediction models of machine learning algorithm replacement strategy. When comparing the predictions using a single feature and dual features, the main metrics used were the area under the receiver operating characteristic (ROC) curve (AUC), specificity (SPE) and sensitivity (SEN). Additionally, the model's performance was comprehensively evaluated using mean absolute error (MAE) and F1 score. To evaluate the overall dataset, we employed the Mann-Whitney U test to assess the differences in ROC curves between various modal fusion models and single-modal models. Specially, in the prediction model that combines DTI and clinical data, a p-value of approximately $9.183 \times 10^{-4}$ was obtained, which is <0.05 and confirming the statistical significance of the model. When compared with different machine learning algorithm replacement strategies, the main metrics used were AUC under the ROC curve and accuracy (ACC), calculated using an 8-fold cross-validation method. Moreover, the model's performance was comprehensively evaluated using root mean square error (RMSE), MAE, and F1 score.

## Results

In our approach, the classification step primarily serves to transform predicted probabilities into labels, considering the two possibilities of disease progression: improvement and deterioration. Consequently, classification is carried out for these two categories. We analyze the experimental results from three key perspectives: (1) comparison of model performance with versus without CFM; (2) evaluation of different fusion strategies; and (3) comparison with existing PD progression methods.

### Performance with and without CMF integration

Using a single or dual modality of data to predict PD progression leads to different results. Table 2 presents the average AUC values for the separate prediction of PD progression using each of the three modalities separately. It can be seen that each AUC value is relatively low. In $\Gamma_i$, where $i$ represents the number of characteristics that are randomly combined. The specific combination of characteristics is shown in Table 3. $\Gamma_{21}$, $\Gamma_{23}$, and $\Gamma_{50}$ represent all combinations of features for each modality. It is noteworthy that the AUC values of the single-modality baselines are close to or even slightly lower than the level of random guessing. This highlights the limitations of single data sources in characterizing complex neurological diseases, as the information they provide may be insufficient to construct effective predictive models.

Using our proposed cross-modal fusion prediction method (CMFP), we tested the clinical combination with DTI and DAT, and the AUC results are shown in Table 4. The yellow background area represents the AUC value predicted by the combination of clinical and DTI data, while the purple area represents the AUC value predicted by the combination of clinical and DAT data. The AUC values vary in different quantity feature selection. When selecting 4 clinical characteristics and 7 DTI characteristics, the results were relatively high, reaching 0.7791. Therefore, in the subsequent analysis, we will

**Table 2. The average AUC values for the separate prediction of PD progression using three (Clinical, DTI, and DAT) modalities.** $\Gamma$ represents feature selection, $\Gamma_i$ represents feature combinations, and $i$ represents the number of feature combination. Different data has different feature combinations, as shown in Table 3

| | $\Gamma_2$ | $\Gamma_3$ | $\Gamma_4$ | $\Gamma_5$ | $\Gamma_6$ | $\Gamma_7$ | $\Gamma_{21}$ | $\Gamma_{23}$ | $\Gamma_{50}$ |
|---|---|---|---|---|---|---|---|---|---|
| **Clinical** | 0.4077 | 0.4371 | 0.4515 | 0.4478 | - | - | - | 0.4629 | - |
| **DTI** | 0.4208 | 0.4476 | 0.4489 | 0.4310 | 0.4770 | 0.4191 | - | - | 0.4381 |
| **DAT** | 0.5017 | 0.4767 | 0.4631 | - | - | - | 0.4506 | - | - |

**Table 3. $\delta_i$, $\gamma_i$ and $\vartheta_i$ represent feature combinations for clinical, DTI, and DAT data, respectively, where $i$ represents the number of feature combinations.** UP3: UPDRS Part III score, PTs: UPDRS Total Score, Uts: UPSIT score, STs: STAI scores. SCC: Splenium of Corpus Callosum, Fcb: Fornix (Column and Body of Fornix), ICl: Inferior Cerebellar Peduncle (Left), SCl: Superior Cerebellar Peduncle (Left), FcSr: Fornix (Crescent)/Stria Terminalis (Right), TaR: Tapetum Right, TaL: Tapetum Left. PuL: Putamen Left, HPu: High Putamen, LSt: Low Striatum, HSt: High Striatum.

| | | | | | | | |
|---|---|---|---|---|---|---|---|
| $\delta_2$ | Age | UP3 | | | | | |
| $\delta_3$ | Age | UP3 | PTs | | | | |
| $\delta_4$ | Age | UP3 | PTs | Uts | | | |
| $\delta_5$ | Age | UP3 | PTs | Uts | STs | | |
| $\gamma_2$ | SCC | Fcb | | | | | |
| $\gamma_3$ | SCC | Fcb | ICl | | | | |
| $\gamma_4$ | SCC | Fcb | ICl | SCl | | | |
| $\gamma_5$ | SCC | Fcb | ICl | SCl | FcSr | | |
| $\gamma_6$ | SCC | Fcb | ICl | SCl | FcSr | TaR | |
| $\gamma_7$ | SCC | Fcb | ICl | SCl | FcSr | TaR | TaL |
| $\vartheta_2$ | PuL | HPu | | | | | |
| $\vartheta_3$ | PuL | HPu | LSt | | | | |
| $\vartheta_4$ | PuL | HPu | LSt | HSt | | | |

Table 4. **The comparison of AUC values for the separate predictions combining clinical data with DTI and DAT.** $\delta_i$, $\gamma_i$, and $\vartheta_i$ represent feature combinations for clinical, DTI, and DAT data, respectively. $\delta_i$, $\gamma_i$, and $\vartheta_i$ represent the same as defined in Table 3. A: denotes the combination prediction of clinical and DTI, B: denotes the combination prediction of clinical and DAT.

| A | $\delta_{23}$ | $\delta_2$ | $\delta_3$ | $\delta_4$ | $\delta_5$ |
|---|---|---|---|---|---|
| $\gamma_{50}$ | 0.6409 | 0.4647 | 0.5539 | 0.4873 | 0.4859 |
| $\gamma_2$ | 0.4955 | 0.5113 | 0.5506 | 0.5102 | 0.6459 |
| $\gamma_3$ | 0.5315 | 0.5161 | 0.5053 | 0.5193 | 0.5858 |
| $\gamma_4$ | 0.5458 | 0.4963 | 0.5435 | 0.4566 | 0.6038 |
| $\gamma_5$ | 0.5288 | 0.4936 | 0.4623 | 0.6019 | 0.4905 |
| $\gamma_6$ | 0.4872 | 0.6439 | 0.7096 | 0.5681 | 0.6135 |
| $\gamma_7$ | 0.6677 | 0.5228 | 0.4717 | **0.7791** | 0.5172 |
| B | $\delta_{23}$ | $\delta_2$ | $\delta_3$ | $\delta_4$ | $\delta_5$ |
| $\vartheta_{21}$ | 0.5759 | 0.4670 | 0.4144 | 0.4606 | 0.4960 |
| $\vartheta_2$ | 0.5592 | **0.6000** | 0.5683 | 0.5694 | 0.5842 |
| $\vartheta_3$ | 0.5591 | 0.5990 | 0.4982 | 0.4657 | 0.5146 |
| $\vartheta_4$ | 0.5729 | 0.5207 | 0.5518 | 0.5130 | 0.5882 |

use this combination as the experimental object. However, the combined prediction effect of clinical and DAT was relatively poor. When selecting 2 clinical characteristics and DAT characteristics for prediction, the AUC was the highest, at 0.6. Using the method we proposed, the variance after predicting the combination of four clinical characteristics and seven DTI characteristics eight times is 0.1085. Meanwhile, after predicting the combination of two clinical characteristics and two DAT characteristics eight times, the variance is 0.1929. Relatively speaking, the combined prediction effect of clinical with DTI was better than that of clinical with DAT.

In Table 4, it can be seen that when combining the four characteristics of clinical data with the seven characteristics of DTI data for prediction, the AUC value obtained is the best among all the combinations of characteristics. Similarly, when the two characteristics of the clinical data are combined with the two characteristics of the DAT data for prediction, a better AUC value is also obtained. To further analyse the performance of the single-modality data prediction model and the dual-modality data prediction model, we calculated the MAE and the F1 score. Fig 2(a) and 2(b) respectively display the MAE and F1 Score evaluation results for the five cases of predictions. MAE is used to measure the average deviation between the predicted results and the actual observed values. A smaller MAE value indicates that the model's predicted results are more consistent with the true values. F1 Score is a harmonic mean based on precision and recall, where a higher value indicates a better classification performance of the model.

In Fig 2(a), it can be seen that among the three single-modality data predictions, the use of DTI data achieves the best MAE performance. Among the two dual-modality fusion predictions, the combination of clinical data and DTI data has a smaller MAE. Furthermore, in Fig 2(b), it can be observed that of the five prediction results, the combination of clinical data and DTI data has the highest F1 score. At the same time, the F1 score for single-modality prediction is much lower than that for dual-modality prediction. We also calculated the variances of the different predictions in terms of the MAE and F1 Score metrics, as shown in Fig 2(c).

## Evaluation of different fusion strategy

In our proposed model, we incorporated the use of Adaboost. In Fig 3, we present a comparison of the performance of our method with other prominent ML techniques (Logistic Regression (LR), Gaussian Naive Bayes (GaussianNB), Decision Tree (DT), Support Vector Machine (SVM), K-Nearest Neighbours (KNN), Random Forest (RF), Extra Trees (ExtraTree)). In the Fig 3, CMFP refers to the results obtained by the method proposed in this work. To comprehensively assess the performance of the model, we used four key performance metrics: ACC, RMSE, and MAE. ACC represents the ratio of correctly classified samples to the total number of samples. The closer the value is to 1, the higher the accuracy of the model predictions. RMSE measures the gap between predicted and actual values, with smaller values indicating a better

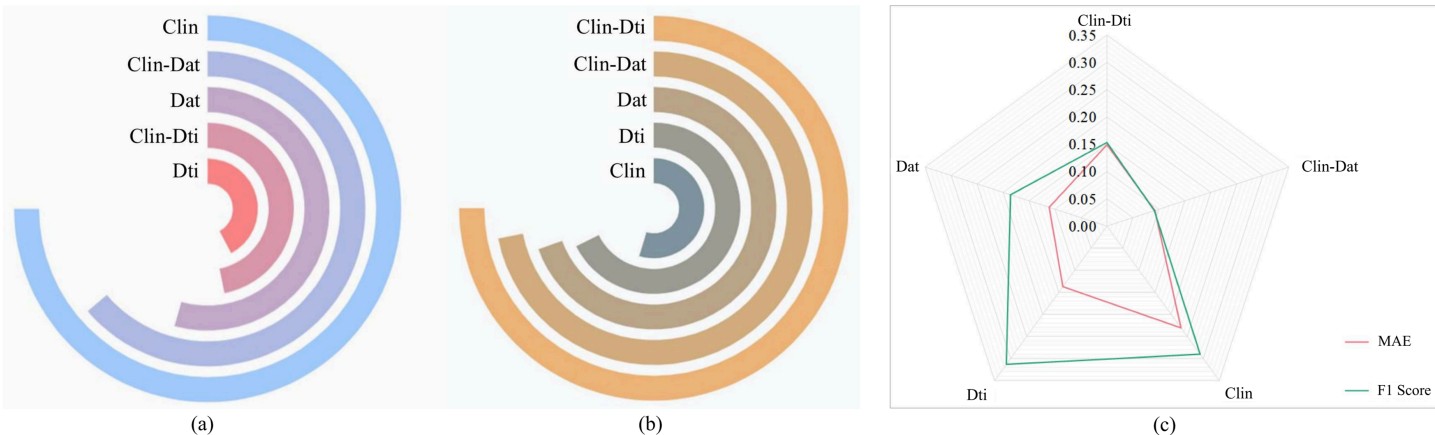

**Fig 2**. The average and variance of MAE and F1 Score values for five predictions of PD progression, utilizing either single-modal or dual-modal data combinations.

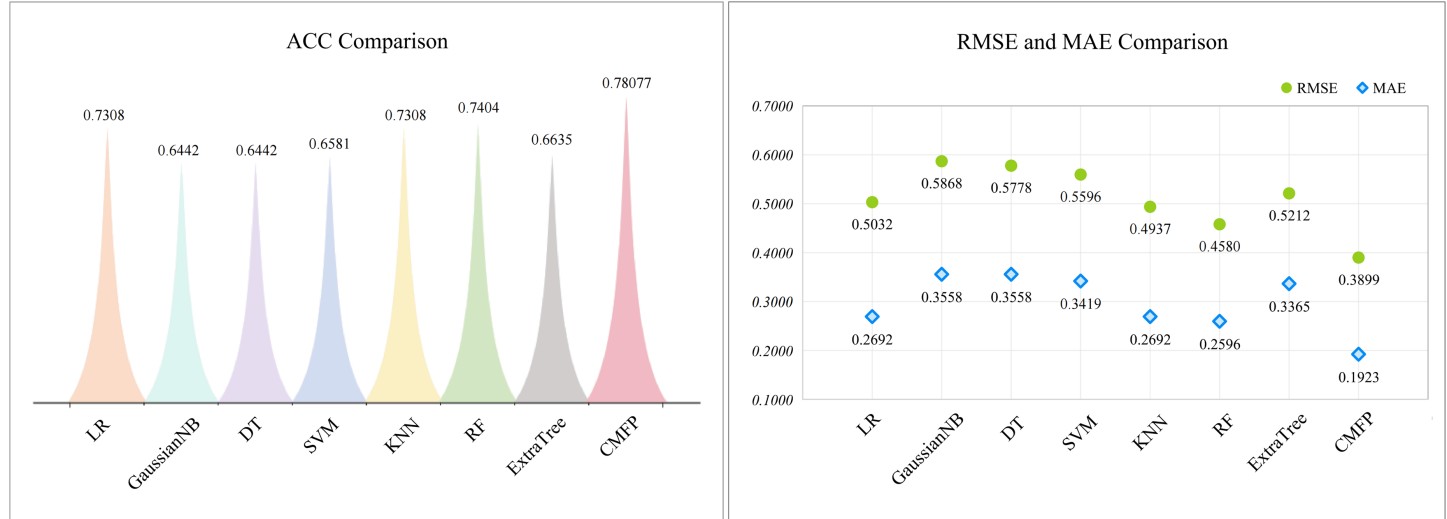

**Fig 3**. By replacing Adaboost with other machine learning methods and calculating four evaluation metrics, we compared the performance of the model.

model fit. MAE measures the average absolute difference between predicted and actual values, and smaller values signify better model performance. From the Figs 3 to 5, it is evident that our method demonstrates significant advantages across all four metrics. Especially in terms of ACC, our method outperforms the second-best RF algorithm by 6.73%. Furthermore, considering the numerical values of RMSE and MAE, compared to RF our algorithm is lower by 6.81% and 6.73%, respectively. These findings further confirm the outstanding performance of our method. In conclusion, our algorithm indeed exhibits remarkable superiority in predictive performance.

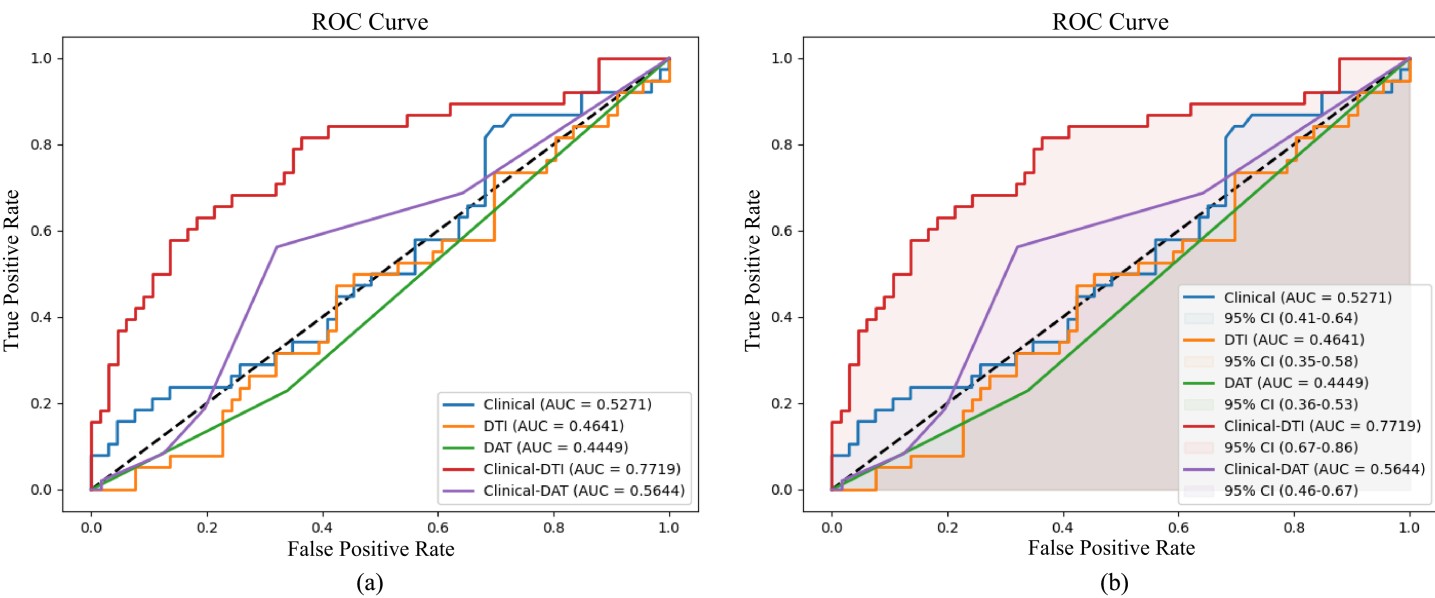

**Fig 4**. The comparison of different ROC in five cases. The shaded area in (b) represents the confidence interval.

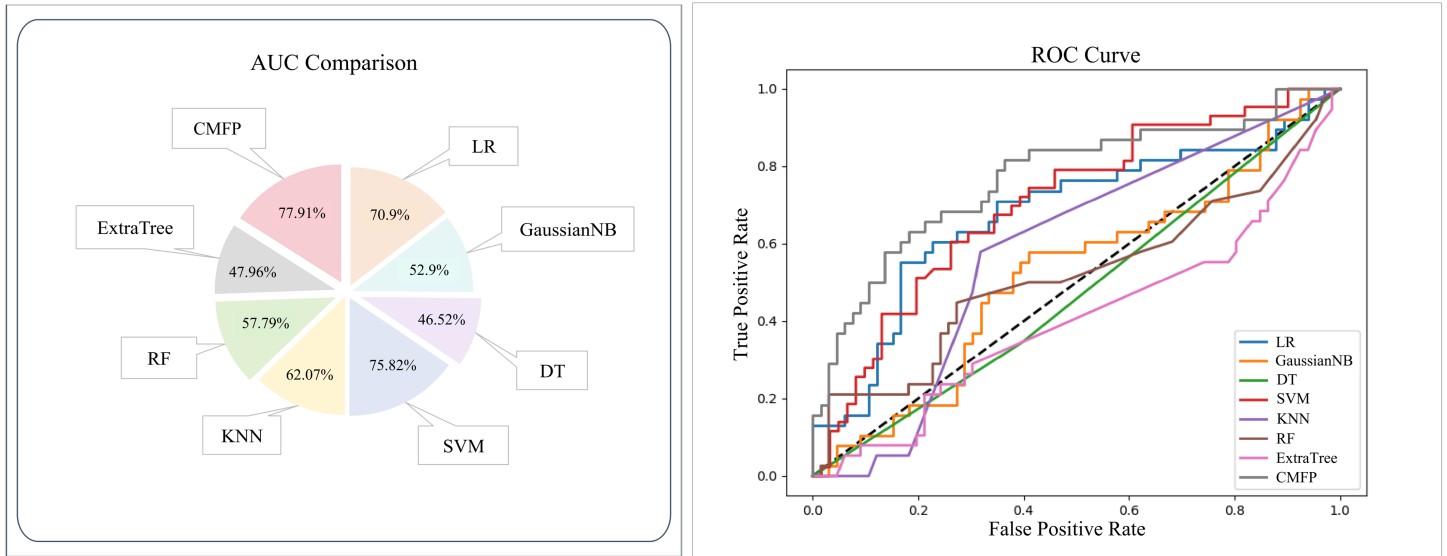

**Fig 5**. By replacing Adaboost with other machine learning methods and calculating four evaluation metrics, we compared the performance of the model.

## Comparison with existing methods

In recent years, numerous novel methods for predicting PD progression have emerged (as summarised in Table 5), with particularly notable advances in multimodal data fusion. Table 5 categorises these approaches into four types: clinical-genetic fusion, clinical-imaging fusion, clinical-genetic-imaging fusion, and clinical-biomarker-imaging fusion. Most studies used the publicly available PPMI dataset and traditional machine learning methods remained dominant in methodology. Among clinical-genetic fusion studies, the work of Chen et al. [24] achieved remarkable performance, which was

**Table 5. Comparison of methods for predicting PD progress using multimodal data fusion.** C-G: clinical and genetic, C-N: clinical and neuroimaging, C-G-N: clinical and genetic and neuroimaging, C-B-N: Clinical and biomarker and neuroimaging.

| Fusion type | Ref. | Dataset: Subjects | Method | Time point | Performance |
|---|---|---|---|---|---|
| C-G | Redenšek et al. [26] | Recruited participants: 220 PD | Linear regression | 5 year | AUC: 0.71 |
| | Chen et al. [24] | PPMI: 409 PD | Logistic regression | 5 year | AUC: 0.8 |
| | Krishnagopal et al. [27] | PPMI: 194 PD | Network-based Trajectory Profile Clustering algorithm | 4 year | ACC: 0.72 |
| C-N | Jackson et al. [28] | PPMI: 139 PD | Logistic, Ridge regression | 1 year | AUC: 0.62 |
| | Tang et al. [29] | PPMI: 69 PD | ANNs | 4 year | ACC: 0.75 |
| | Salmanpour et al. [30] | PPMI: 885 PD | HMLS | 4 year | ACC: 0.792 |
| | Hu et al. [31] | UK Biobank vision cohort: 66500 participants | Logistic regression | 5 year | AUC: 0.717 |
| | Liu et al. [25] | Ruijin Hospital Affiliated to Shanghai Jiao Tong University: 33 PD | Logistic regression | 6 months | ACC: 0.8, AUC: 0.85 |
| | **Ours** | PPMI: 123 PD | **CMFP** | 5 year | AUC: 0.7791, ACC: 0.8077 |
| C-G-N | Sadaei et al. [32] | PPMI, PDBP: 529 PD, 350 PD | XGBoost | 1,2 and 3 year | AUC: 0.77, 0.76 |
| C-B-N | Kim et al. [33] | PPMI: 393 PD | Linear regression | 4 year | AUC: 0.755 |
| | Li et al. [34] | PPMI: 73 HC, 158 PD | LSVM, KNN, Bayes, LDA, Elastic Net | 5 year | AUC: 0.77, ACC: 0.78 |
| | Chen et al. [35] | PPMI: 338 PD | Cox regression | 5 year | AUC: 0.77 |

supported by a large sample size to enhance statistical power and generalizability. It should be noted, however, that sample size alone does not determine predictive accuracy, as demonstrated by Liu et al. [25]. In clinical-imaging fusion, reference [25] reported better results, but their study used a private dataset, which may limit generalisability. In contrast, the method proposed in this study demonstrates the best performance in predicting 5-year progression, primarily through the fusion of clinical and neuroimaging data. The combination of clinical data and DTI yielded the most significant improvements, which will be discussed further in the following chapter.

## Discussion

### Clinical significance analysis of data

The clinical significance of HYS changes is highly dependent on the specific stage in which they take place. Among them, the transition from Stage 2 to Stage 3 of HYS is of unique and paramount importance, acting as a pivotal clinical milestone in the progression of the disease [36]. This shift typically foreshadows more severe disability, a significantly elevated risk of falls, and alterations in treatment response for patients. In contrast, while changes from stage 1 to stage 2 of HYS indicate progression of symptoms, their impact on patient functional status and prognosis is generally less pronounced compared to the transition from Stage 2 to Stage 3. In addition, this change often occurs during the relatively early "honeymoon period" of the disease. Consequently, if a research cohort comprises a higher proportion of patients experiencing early stage (stage 1-2) changes, the "clinical weight" of their disease progression may be lower than that of a cohort with a greater number of patients undergoing Stage 2-3 transitions. Given these factors, this study grouped patients solely on the basis of whether there was progression of HYS (yes/no), without further distinguishing the specific stage of HYS at which progression occurred.

The HYS scale has certain limitations, with relatively low resolution. In particular, it is difficult for HYS Stage II to accurately capture subtle changes in motor function. In contrast, the MDS-UPDRS Part III score, as a continuous variable scale, can conduct quantitative assessments of specific motor signs (such as tremor, rigidity, bradykinesia, etc.), thus providing a more sensitive and objective means of monitoring the progression of the disease [37]. Given that this study focuses on the critical transition in disease staging (such as the important turning point from Stage II to Stage III), and the HYS scale, as an authoritative standard for clinical staging, has a stronger correlation with the long-term prognosis while

also allowing standardised data collection in multicenter studies [36]. Therefore, it is entirely reasonable for this study to select the HYS scale.

DTI is a new MRI technique developed in recent years. It can use multiple diffusion-sensitive gradients of different sizes to show the strength of the diffusion capacity of water molecules in vivo and the direction of the diffusion movement. It focuses on observing the biochemical composition, microstructure and arrangement of tissues. This technique is the only imaging method to show the non-invasive diffusion characteristics of water molecules in living tissue. DTI can elucidate the pathological changes in the microstructure of cerebellar white matter in patients with PD. This imaging technique not only sensitively captures the patterns of damage to cerebellar white matter during the course of the disease, but also reveals deep associations with key clinical features of PD [6]. Additionally, DAT imaging visually displays the spatial distribution and functional status of dopamine transporters in the brain, directly reflecting the integrity of the nigrostriatal dopaminergic system. Given that dopaminergic dysfunction is the most central pathological feature of PD, the quantitative measures provided by DAT exhibit a strong correlation with the severity and progression stages of the disease. Structural characteristics derived from 3D T1-weighted imaging, such as brain volume and cortical thickness, also hold significant value in Parkinson's disease research. However, in the early stages of PD, structural changes typically emerge later than the microstructural alterations of white matter detected by DTI and the functional abnormalities revealed by DAT. Therefore, in this study, we did not include 3D T1-weighted imaging in our analysis.

Some researchers explored the change in white matter based on the DTI study and found that compared to the control group, white matter damage in the frontal lobe of patients with PD-MCI was more significant and related to damage to general cognitive function and multiple cognitive domains [38,39]. For the complex and diverse neurological mechanisms of PD, multimodal magnetic resonance characteristics are integrated to perform a comprehensive evaluation and analysis from multiple levels and different perspectives, providing a more reasonable method to further improve the precision of the diagnosis of PD [40–42]. Pyatigorskaya et al. [40] used a combination of the volume of the nigrae and the intensity characteristics of the signal of NMS-MRI and the partial anisotropy characteristics in DTI modes to achieve a diagnostic precision of 93% for PD. Lei et al. [43,44] created a PD classification prediction model based on multimodal medical imaging technology (mainly including MRI and DTI) to detect PD intelligently and perform clinical score prediction. Chougar et al. [45] used the volume and DTI of 13 brain regions as input layers, and used a supervised ML algorithm to accurately predict the classification of PD, PSP, and MSA-P. Kim et al. [46] focused their research on newly diagnosed PD patients, comparing them with healthy individuals of the same age. By computing and contrasting quantitative anisotropy values across the subcortical and cortical regions in both cohorts, the aim was to identify the areas of the brain most prominently affected during the early stages of PD. Shin et al. [47] studied the thickness of the MRI cortical to predict the transition from mild cognitive impairment to dementia in PD. This research demonstrated that magnetic resonance cortical thickness helps predict the transition from mild cognitive impairment to dementia in PD at the individual level, with better performance when combined with clinical data.

Combining the results of Tables 2 and 4, key clinical metrics that contribute to the prediction of PD can be identified, leading to inference of potential characteristics associated with the progression of PD. Among these, crucial clinical data characteristics include age, UPDRS Part III Score, UPDRS Total Score, and UPSIT score, as shown in Table 1.

Recently, the advent of reliable network characterisation techniques has made it possible to understand neurological disorders at the level of total brain connectivity. However, so far, few studies have used white matter data as the main object to predict the progression of PD. Wee et al. [48] proposed an effective web-based multivariate classification algorithm that uses white matter fiber data and accurately identifies patients with MCI from normal controls. The results suggest that the proposed classification framework could provide an alternative and complementary approach to the clinical diagnosis of brain structural changes associated with cognitive impairment, but this research was focused on Alzheimer's disease. Huang et al. [7] adopted the elastomere consensus ranking (ENFCR) method based on networks to explore the potential of the baseline features of the structural connectivity of white matter obtained through DTI to predict the future development of MCI in newly diagnosed PD patients, which indicated that the structural connectivity of white matter plays

a significant role in predicting the progression from PD to MCI. However, no significant biomarkers were identified. Zhang et al. [49] used the TBSS method to analyse FA of brain white matter (WM) in patients with PD versus healthy controls, finding that integrating WM lesion regions with clinical information significantly improves prediction accuracy for disease progression. However, this method requires manual intervention and there is uncertainty in the location of WM-lesion areas.

### Advantages of cross modal data fusion prediction

In the model of this paper, aside from clinical features, seven features(Splenium of Corpus Callosum & Fornix (Column and Body of Fornix) & Inferior Cerebellar Peduncle (Left) & Superior Cerebellar Peduncle (Left) & Fornix (Crescent)/Stria Terminalis (Right) & Tapetum Right & Tapetum Left) were selected from DTI variables for brain white matter MD (50) data, as shown in Table 1. White matter at the individual level contributes to the prediction of PD progression and, when combined with clinical characteristics, improves the predictive performance of the model. The predictive model based on DTI's global white matter features has an AUC of 0.44035 (variance is 0.0185), the model based solely on clinical features has an AUC of 0.4414 (variance is 0.0187), and the predictive model that combines both DTI's white matter features and clinical features has an AUC of 0.7791 (variance is 0.1085).

Compared to single-modal approaches, multimodal fusion techniques can achieve information complementarity, capturing a more complete representation of brain changes in patients with PD. This allows for a holistic evaluation and analysis of the disease from various points of view, providing crucial information for a comprehensive diagnostic assessment.

Fig 4 displays five ROC curves, corresponding to five different scenarios: using clinical data alone, DTI data alone, DAT data alone, combining clinical and DTI data, and combining clinical and DAT data. In the figure, five distinct colours represent the prediction results obtained from these five different data scenarios. In Fig 4(a), based on the positional range of the ROC curves, the AUC values can be determined. It can be observed that when clinical, DTI, or DAT data are used individually, AUC values are relatively low. However, when cross-modal fusion of two types of image data (DTI/DAT) with clinical data is used to predict progression of PD, the predictive performance of single-modal methods can be improved. Upon comparison, it is found that the combination of clinical and DTI data yields an even better prediction, with a significant increase in the AUC value. In Fig 4(b), the shaded area represents the confidence interval. It can be seen that the size of the confidence interval is also positively correlated with the AUC value. There is a 95% probability that the overall parameter for the prediction combining clinical and DTI data falls within this range, indicating that DTI data are more beneficial than DAT data in predicting the progression of PD.

Furthermore, as indicated in Tables 2, 4 and Fig 4, combining clinical data with DTI data for prediction performs better than combining clinical data with DAT data. We also performed a Mann-Whitney U test, which revealed that the combined prediction of clinical and DTI data is statistically significant compared to using clinical data alone, with a p-value of 9.183e-4.

To further compare the combined prediction results of clinical data with DTI and DAT, we conducted a comparative analysis by calculating the metrics ACC, Sensitivity, and Specificity, as shown in Table 6. The data in the table represent the average and variance values after eight tests, and it is evident that the combination of clinical and DTI consistently yields relatively higher values in all metrics. As indicated by the variance of each metric in the Table 6, the current model performance still exhibits certain fluctuations and requires further improvement. This performance variance reflects the uncertainty of model predictions under conditions of limited and heterogeneous data. Nevertheless, the core conclusions of this study remain robust, supported by statistical significance and consistent performance ranking. Future work should involve larger-scale, multi-center data to further validate the stability and reproducibility of the model.

In summary, combined prediction of clinical and DTI data is more advantageous in predicting the progression of PD.

**Table 6. The comparison of the average (Avg) and variance (Vac) values for the four metrics in separate predictions combining clinical data with DTI and DAT.** Clin-Dti: denotes the combination prediction of clinical and DTI, Clin-Dat: denotes the combination prediction of clinical and DAT.

| *Avg* | AUC | ACC | Sensitivity | Specificity |
|---|---|---|---|---|
| Clin-Dti | **0.7791** | **0.8077** | **0.9464** | **0.7470** |
| Clin-Dat | 0.6000 | 0.7385 | 0.8679 | 0.6298 |
| *Vac* | AUC | ACC | Sensitivity | Specificity |
| Clin-Dti | **0.1085** | 0.1490 | 0.1417 | 0.1890 |
| Clin-Dat | 0.1929 | **0.0923** | **0.1041** | **0.1516** |

In order to demonstrate the superiority of our approach, we further analyse the AUC values under different scenarios of replacing machine learning methods. The AUC metric was employed to quantitatively assess the model's discriminative performance, with values ranging from 0 to 1, with higher values indicating better model performance.

Fig 5 presents a comparison of the predictive performance of different fusion algorithms, the left side showing a comparison of AUC values and the right side showing the corresponding distributions of the ROC curves. Among the several machine learning algorithms listed, it can be observed that our method achieves higher AUC values and demonstrates better performance in terms of the ROC curves. Specifically, in Fig 5, although SVM ranks as the second-best algorithm, our method still outperforms it by 2.09%. In fact, our algorithm exhibits significant superiority in predictive performance. In Fig 5, CMFP refers to the results obtained by the method proposed in this work. Here, despite that SVM has the second highest AUC value among algorithms, our method still exceeds it by 2.09%.

DTI provides us with a unique perspective to gain insight into the microscopic structural changes of living tissues. Among them, the analysis of DTI along perivascular spaces (DTI-ALPS) metric has emerged in recent years as a promising method to assess human lymphatic system function, attracting widespread attention from academic and clinical fields [50]. The DTI-ALPS is derived by calculating the diffusion ratio along perivascular spaces (PVS) in the periventricular white matter, and this metric is regarded an indirect measure of lymphatic system status and function. Not only does it offer a new perspective for understanding the operating mechanisms of the lymphatic system, but it also provides a powerful tool for assessing the progression of neurodegenerative diseases and other neurological disorders.

The concept of DTI-ALPS was initially proposed by Taoka [51] in the research on Alzheimer's disease. The research showed a significant positive correlation between ALPS along perivascular spaces and MMSE scores, implying that as the severity of AD increases, the rate of water diffusion along perivascular spaces decreases. This discovery laid a solid foundation for subsequent research in fields such as PD. In PD, DTI-ALPS has also demonstrated its significant value, with PD patients showing a markedly lower ALPS metric compared to healthy controls [50,52–54]. More importantly, the differences in ALPS also reflect the relationship between lymphatic clearance, cognitive function, and disease severity in patients with PD [54]. Research has found that a lower baseline DTI-ALPS is closely associated with subsequent declines in cognitive ability and worsening of disease severity [54]. Therefore, DTI-ALPS has a considerable reference value in improving the accuracy of PD progression predictions.

To further validate the effectiveness of DTI-ALPS in predicting the progression of PD, we conducted related research. In our research, we calculated the ALPS values for the left cerebral hemisphere of subjects. We combined them with clinical data, using multiple metrics to evaluate the effectiveness of the prediction and the model. As shown in Fig 6(a), among the six metrics, the prediction effect using a combination of clinical data and ALPS was significantly better than using clinical data combined with DTI, with an improvement in ACC of 3.85%. Fig 6(b) presents the performance of a further analysis model using a combination of clinical data and ALPS values. We compared the Adaboost algorithm with other machine learning algorithms and found that our method outperformed others in terms of multiple evaluation metrics. In particular, the F1 score, which balances both precision and recall of classification models, was 1.52% higher for our method compared to the suboptimal method. Meanwhile, in terms of RMSE, our method was also 5.19% lower than the suboptimal algorithm.

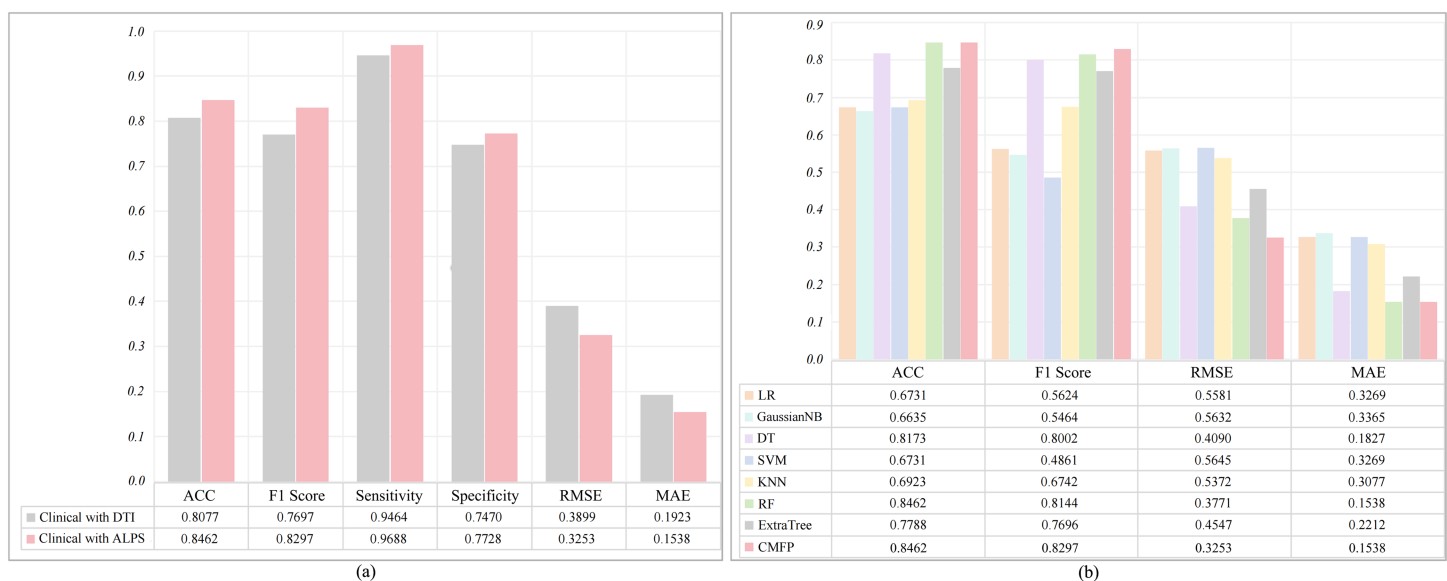

**Fig 6. (a) Clinical+DTI vs. DTI+ALPS comparison; (b) Replace Adaboost with other ML methods for comparison.**

In summary, our research shows that a method that combines DTI white matter data with clinical characteristics helps predict PD progression more accurately. Furthermore, the construction of whole brain white matter features based on machine learning provides a new approach to assess and monitor disease severity in PD patients, helping in early clinical formulation of more effective PD treatment plans. Furthermore, with continued technological advancements and deeper research, methods based on DTI-ALPS will exhibit even broader application prospects in predicting the progression of PD.

## Conclusion

The progression of PD disease is a long and progressive process, and its diagnosis depends mainly on clinical signs and symptoms. It is very difficult to treat the disease in the middle and late stages. Therefore, early prediction of the disease is crucial for the implementation of appropriate intervention measures. However, there is still a lack of reliable biomarkers for the progression of PD, so it is of great significance to establish a model for early prediction of disease progression. DTI can quantitatively measure the extent and direction of the diffusion of water molecules in the brain and assess the structural integrity and continuity of the brain's white matter fibres so that it can indicate potential changes in the early stage of the disease. The method in this paper is constructed based on longitudinal data research, and the ML method is used to establish a disease progression model of PD by integrating baseline DTI and clinical characteristic data of early PD, which is expected to provide an imaging basis for early decision-making of PD.

This paper is based mainly on clinical data and DTI/DAT cross-modal fusion to predict PD progression. The main limitation is that this research has only been validated in public data sets and lacks external data validation. Although HYS changes vary in clinical significance in stages, this study did not distinguish the exact stage of progression. Although the selection of the HYS scale as a marker of disease progression in this study offers the advantage of universal applicability in clinical staging, its insufficient resolution may weaken the analysis of the correlation with continuous changes in motor function. Future studies should combine fine-grained scales such as MDS-UPDRS to validate the generalisability of this classifier. Additionally, data validation for the model was not performed using datasets from other sources. Our further

**Table 7. Abbreviation and specific full name.**

| Abbreviation | Specific name | Abbreviation | Specific name |
|---|---|---|---|
| PD | Parkinson's disease | CMFP | Cross-Modal Fusion Prediction Model |
| CMF | Cross-Modal Fusion | DTI | Diffusion tensor imaging |
| DAT | Dopamine transporter | ALPS | Along the perivascular space |
| FA | Fractional Anisotropy | DTI-ALPS | the DTI metric along the perivascular space |
| MD | Mean Diffusivity | UPDRS | Unified Parkinson's Disease Rating Scale |
| ML | Machine learning | MDS-UPDRS | Movement Disorder Society-sponsored revision of the UPDRS |
| HYS | Hoehn and Yahr Scale | PPMI | Parkinson's Progression Markers Initiative |
| MRI | Magnetic Resonance Imaging | FACT | Fiber Assignment by Continuous Tracking |
| MSE | Mean Squared Error | ESS | Epworth Sleepiness Scale |
| GDS | Geriatric Depression Scale | UPSIT | University of Pennsylvania Smell Identification Test |
| MoCA | Montreal Cognitive Assessment | RBDSQ | Rapid Eye Movement Sleep Behavior Disorder Screening Questionnaire |
| CSF | Cerebrospinal Fluid | ROC | Receiver Operating Characteristic Curve |
| AUC | Area Under the Curve | SPE | Specificity |
| SEN | Sensitivity | ACC | Accuracy |
| MAE | Mean Absolute Error | RMSE | Root Mean Square Error |
| LR | Logistic Regression | GaussianNB | Gaussian Naive Bayes |
| DT | Decision Tree | SVM | Support Vector Machine |
| KNN | K-Nearest Neighbours | RF | Random Forest |
| ExtraTree | Extra Trees | NMS-MRI | Neuromelanin-Sensitive Magnetic Resonance Imaging |
| PSP | Progressive Supranuclear Palsy | MSA-P | Multiple System Atrophy with Predominant Parkinsonism |
| ENFCR | Elastomere Consensus Ranking | MCI | Mild Cognitive Impairment |
| TBSS | Tract-Based Spatial Statistics | WM | White Matter |
| PVS | Perivascular Spaces | | |

research will combine multimodal data, such as grey matter structure data and gene data, to further study PD and other neurological diseases.

In this paper, numerous abbreviations are used, and their specific meanings can be found in the Table 7.

## Author contributions

**Data curation:** Amei Chen, Xinhua Wei.

**Formal analysis:** Amei Chen, Hua Xiong, Xinhua Wei.

**Investigation:** Jinyu Wen, Hua Xiong, Xinhua Wei.

**Methodology:** Jinyu Wen, Meie Fang.

**Validation:** Jinyu Wen, Amei Chen, Jingxin Liu, Meie Fang.

**Writing – original draft:** Jinyu Wen, Meie Fang.

**Writing – review & editing:** Jinyu Wen, Jingxin Liu, Hua Xiong, Xinhua Wei.

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
