## [Decision Letter · Decision Letter 0]

25 Jun 2025

PONE-D-25-29378Cross-modal fusion of brain imaging and clinical data for Parkinson's disease progression predictionPLOS ONE

Dear Dr. Wen,

Thank you for submitting your manuscript to PLOS ONE. After careful consideration, we feel that it has merit but does not fully meet PLOS ONE’s publication criteria as it currently stands. Therefore, we invite you to submit a revised version of the manuscript that addresses the points raised during the review process.

The reviewers' comments for the authors are provided below. If there are any comments you are unable or choose not to address, please include an explanation. While it is not mandatory to implement every suggestion, the feedback from the reviewers and editor is intended to help improve the overall quality of your manuscript and should be carefully considered. We would be pleased to reconsider your manuscript should you choose to submit a revised version.

We look forward to receiving your revised manuscript.

Kind regards,

Nima Broomand Lomer, M.D.

Academic Editor

PLOS ONE

Journal Requirements:

“This work was supported in part by the National Natural Science Foundation of China (No. 62072126), in part by the Fundamental Research Projects Jointly Funded by Guangzhou Council and Municipal Universities No. SL2023A03J00639, in part by the Key Laboratory of Philosophy and Social Sciences in Guangdong Province of Maritime Silk Road of Guangzhou University (GD22TWCXGC15), in part by the Natural Science Foundation of Chongqing (No. CSTB2024NSCQ-MSX1087), and the Guangxi Science and Technology Program (No.AD23023001).”

5. Thank you for stating in your Funding Statement:

“This work was supported in part by the National Natural Science Foundation of China (No. 62072126), in part by the Fundamental Research Projects Jointly Funded by Guangzhou Council and Municipal Universities No. SL2023A03J00639, in part by the Key Laboratory of Philosophy and Social Sciences in Guangdong Province of Maritime Silk Road of Guangzhou University (GD22TWCXGC15), in part by the Natural Science Foundation of Chongqing (No. CSTB2024NSCQ-MSX1087), and the Guangxi Science and Technology Program (No.AD23023001).”

“This work was supported in part by the National Natural Science Foundation of China (No. 62072126), in part by the Fundamental Research Projects Jointly Funded by Guangzhou Council and Municipal Universities No. SL2023A03J00639, in part by the Key Laboratory of Philosophy and Social Sciences in Guangdong Province of Maritime Silk Road of Guangzhou University (GD22TWCXGC15), in part by the Natural Science Foundation of Chongqing (No. CSTB2024NSCQ-MSX1087), and the Guangxi Science and Technology Program (No.AD23023001).”

“This work was supported in part by the National Natural Science Foundation of China (No. 62072126), in part by the Fundamental Research Projects Jointly Funded by Guangzhou Council and Municipal Universities No. SL2023A03J00639, in part by the Key Laboratory of Philosophy and Social Sciences in Guangdong Province of Maritime Silk Road of Guangzhou University (GD22TWCXGC15), in part by the Natural Science Foundation of Chongqing (No. CSTB2024NSCQ-MSX1087), and the Guangxi Science and Technology Program (No.AD23023001).”

7. Please include a separate caption for each figure in your manuscript.

Reviewers' comments:

Reviewer's Responses to Questions

**Comments to the Author**

1. Is the manuscript technically sound, and do the data support the conclusions?

Reviewer #1: Yes

Reviewer #2: Partly

Reviewer #3: No

2. Has the statistical analysis been performed appropriately and rigorously?

Reviewer #1: Yes

Reviewer #2: No

Reviewer #3: No

3. Have the authors made all data underlying the findings in their manuscript fully available?

Reviewer #1: Yes

Reviewer #2: Yes

Reviewer #3: Yes

4. Is the manuscript presented in an intelligible fashion and written in standard English?

Reviewer #1: Yes

Reviewer #2: No

Reviewer #3: No

5. Review Comments to the Author

**Reviewer #1:** 1.Clinical Information of Study Subjects

Please provide a demographic table summarizing the clinical characteristics of the Parkinson’s disease patients included in this study. In particular, information for both the HYS deterioration group and the non-deterioration group at baseline and at the 5-year follow-up is necessary to allow for a clearer understanding of cohort composition and comparability.

2. Clinical Significance of HYS Changes at Different Stages

This study categorizes patients based on the presence or absence of Hoehn and Yahr Scale (HYS) progression. However, the clinical significance of changes in HYS scores depends heavily on the specific stages involved. For example, a change from stage 1 to 2 may simply reflect a transition within the early, so-called “honeymoon” phase, while a shift from stage 2 to 3 typically indicates entry into the more progressive stage of the disease. I recommend addressing this issue in the discussion or including it as part of the limitations of the study.

3. Limitations of HYS as an Assessment Scale

The HYS is a relatively coarse measure of disease severity. In clinical practice, many PD patients are classified as stage II, but the degree of motor symptom severity within this group can vary widely. Therefore, many previous studies use MDS-UPDRS Part III scores as a more granular and sensitive indicator of motor function. Please consider discussing this limitation of the HYS in light of the metrics used in prior literature, particularly as it relates to your classifier’s outcome and feature selection.

4. Comparison with Previous Studies

While this study proposes a novel machine learning classifier to predict PD progression, there is a large body of existing literature with similar goals. However, the current manuscript lacks a comparative analysis of the proposed model’s performance relative to those of prior studies. I strongly recommend including a discussion of how your model’s AUC and other evaluation metrics compare to previously published classifiers, to contextualize the novelty and value of your findings.

5. Rationale for Feature Selection

Although clinical features were selected using the Lasso method, the imaging modalities (DTI and DAT SPECT) were chosen a priori. The rationale for restricting imaging feature selection to DTI and DAT SPECT is not clearly explained. Given that many previous studies have employed structural features extracted from 3D T1-weighted images—such as brain volume and cortical thickness—it would be worth discussing whether incorporating such features could potentially improve classifier performance.

**Reviewer #2:** The authors aimed to develop a model based on MRI/DTI scans to provide a reliable biomarker useful for prediction of PD at its early stage. With the PPMI database of multi-center PD patients, the proposed cross-modality fusion prediction method (CMFP) appears superior in PD-progression prediction performance compared to single-modality approach. Machine learning is an increasing promising tool for clinic diagnosis; however, the comparative predictive values of its use combined with MRI/DTI remain unclear. In this sense, the manuscript provided a badly needed model to this end. I’ve several major and minor concerns regarding this manuscript (see below).

Major concerns:

1. The authors reported their results in a very casual way. They basically skipped the reporting of the Figure 4 and the Figure 5 in the Results (lines 260), which was then brought into a detailed description in the Discussion. As a result, the Discussion session was mixed with results and discussion;

2. It’s very confusing that it appears the CMFP is the prediction model the authors were trying to sell, yet CMFP was rarely seen throughout the manuscript (not once in the Discussion). In contrast, the authors focused on the Decision Fusion Predictive Model (line 96);

3. Many multi-modal prediction models exist as the author mentioned (refs 25-29). The rationale and mechanisms the CMFP were superior to other models is not clear in this manuscript. Have the authors applied these models on the PPMI dataset used in this study? Have they authors applied their CMFP model to other PD dataset(s)? The comparison was missing in the Results. But at least these should be mentioned in the Discussion.

Minor concerns:

1. The writing and organization of the manuscript is chaotic, specifically: 1) the long passage (lines 16-51) in the Introduction session is better to split into two parts, 2) using the bullet points in the Introduction is discouraged, 3) Python and Pytorch should be in the Methods session (lines 199-200);

2. The English/grammar need to be improved/corrected significantly throughout the manuscript. It’s difficult to read and follow. For example, the result summary titles (the lines 206, 248) were illy composed. In the line 359, “AUC used to measure the model’s classification ability.” And many more throughout the text and legends;

3. The statistically significance level was not specified. What main statistic method was used for group comparison (other than Mann-Whitney U test)? How was the statistics performed? How were the age and sex controlled?

4. In the Table 5, full names of SEN and SPE should be used. No need to abbreviate;

5. Remove the numbering in the Methods and Results sessions;

6. Remove the redundant abbreviation of PD in the line 274;

7. The manuscript images seem fuzzy and hard to read. Were the authors using appropriate dpi?

**Reviewer #3: **Researchers attempted to predict Parkinson's disease progression using three data modalities (clinical, DTI imaging, DaTscan) combined through cross-modal fusion to improve upon single-modality predictions. Although the methodology appears rigorous, fundamental questions remain unanswered.

1. Serious methodological problems:

i. Training for 120 epochs with only 123 total patients (even fewer after data splitting) will inevitably cause the model to memorize training examples rather than learn generalizable patterns. Why was such high epoch used?

ii. How is AdaBoost integrated with Adam optimizer? AdaBoost operates through weighted voting without gradient descent, while Adam is specifically designed for gradient-based neural network optimization - these approaches are fundamentally incompatible.

iii. How is a classification model using mean squared error loss? Classification problems require cross-entropy loss, not regression losses like MSE that are designed for continuous value prediction.

iv. Feature selection performed before data splitting is fundamentally wrong and promotes data leakage, where the selection process "sees" test data and artificially inflates performance estimates.

v. There is a sample bias between positive and the negative class. Was the split stratified?

2. Other problems:

i. Variables in equations 1-6 (ρ, ψ, τ, μ, ϕ, λ, ι, α, υ) lack proper definition - what do these variables represent and how do they relate to the fusion process?

ii. No nested cross-validation, no external validation, confidence intervals missing.

iii. Authors show only one p-value which signifies one statistical test but the authors comment about multiple tests being performed. Where is the multiple comparison test?

iv. Claims such as novel CMF without proper citations. There are multiple claims without citations.

3. Grammar:

i. Poor grammar encountered numerous times in the paper.

ii. They write "using imaging-omics, particularly clinical data" which is contradictory since clinical data typically isn't considered part of imaging-omics.

iii. The authors inconsistently refer to their method as "CMFP," "CMF," and "cross-modal fusion" without clearly establishing these as equivalent terms.

iv. Claims like "DTI metric along the perivascular space has relatively more advantages" is vague without specifying compared to what.

6. PLOS authors have the option to publish the peer review history of their article (what does this mean?). If published, this will include your full peer review and any attached files.

Reviewer #1: **Yes: **kazuhide seo

Reviewer #2: No

Reviewer #3: No

---

## [Author Response · Author response to Decision Letter 1]

23 Jul 2025

The reviewer’s comments and suggestions have been carefully addressed in the revised version of the paper. Changes in the revised version have been colored red. We are grateful once again to the associate editor and reviewers for their time and helpful comments, which have played a vital role in improving the quality and presentation of the original manuscript.

---

## [Decision Letter · Decision Letter 1]

21 Aug 2025

PONE-D-25-29378R1Cross-modal fusion of brain imaging and clinical data for Parkinson's disease progression predictionPLOS ONE

Dear Dr. Wen,

Thank you for submitting your manuscript to PLOS ONE. After careful consideration, we feel that it has merit but does not fully meet PLOS ONE’s publication criteria as it currently stands. Therefore, we invite you to submit a revised version of the manuscript that addresses the points raised during the review process.

Please address the concerns raised by Reviewer 3 thoroughly and resubmit the manuscript for reevaluation. If any comments cannot be addressed, provide a clear justification.

We look forward to receiving your revised manuscript.

Kind regards,

Nima Broomand Lomer, M.D.

Academic Editor

PLOS ONE

Journal Requirements:

Reviewers' comments:

Reviewer's Responses to Questions

**Comments to the Author**

1. If the authors have adequately addressed your comments raised in a previous round of review and you feel that this manuscript is now acceptable for publication, you may indicate that here to bypass the “Comments to the Author” section, enter your conflict of interest statement in the “Confidential to Editor” section, and submit your "Accept" recommendation.

Reviewer #1: All comments have been addressed

Reviewer #2: All comments have been addressed

Reviewer #3: (No Response)

2. Is the manuscript technically sound, and do the data support the conclusions?

Reviewer #1: Yes

Reviewer #2: Yes

Reviewer #3: Partly

3. Has the statistical analysis been performed appropriately and rigorously?

Reviewer #1: Yes

Reviewer #2: Yes

Reviewer #3: No

4. Have the authors made all data underlying the findings in their manuscript fully available?

Reviewer #1: Yes

Reviewer #2: Yes

Reviewer #3: Yes

5. Is the manuscript presented in an intelligible fashion and written in standard English?

Reviewer #1: Yes

Reviewer #2: Yes

Reviewer #3: No

6. Review Comments to the Author

Reviewer #1: I have carefully reviewed the authors' detailed responses and the revised manuscript. Thank you for thoroughly and appropriately addressing all of the reviewer comments I raised. I have no further comments or concerns.

Reviewer #2: The authors have addressed all my concerns. I have no further questions. With that said, the revised Discussion is quite lengthy and the authors should make it more relevant and more precise.

Reviewer #3: Thanks to the authors for the changes. Some of the problems were adequately answered but the paper remains confusing.

Major:

1. Single-modality AUC values (0.40-0.50) are worse than random chance, indicating a methodological issue.

2. Patient number mismatch: "…it was found that 74 patients had scores higher than baseline…" vs "…progression group (n = 72)…".

3. Missing confidence intervals for AUC values in Tables 2, 4, and 6. With reported variance of 0.1085, and without confidence intervals the reliability of the 0.7791 AUC cannot be assessed.

4. High performance variance (0.1085) indicates unstable results but no discussion of reproducibility implications.

Minor:

1. CMFP/CFMP, CMF/CFM used interchangeably.

2. "Multivariate regression analysis further confirmed a negative…" missing a citation.

3. ACC, AUC, ESS, RBDSQ, UPSIT, etc undefined in introduction. Please provide a table for all the abbreviations.

4. In table 2 ' tau i' should explicitly mention that 'i' is just the corresponding 'i' as given in table 3. Otherwise it looks confusing. Or present table 3 before 2.

5. "The clinical significance of HYS…" the first three paragraphs of discussion, present claims without citations.

6. "four clinical data" in methods should be "four clinical variables" and other grammatical errors throughout.

7. "Larger sample size for improved prediction" in existing methods comparison. Larger sample size reduces variance but doesn't guarantee better performance - Liu et al. (citation 25) achieved higher AUC/ACC scores with only 33 PD patients.

7. PLOS authors have the option to publish the peer review history of their article (what does this mean?). If published, this will include your full peer review and any attached files.

Reviewer #1: **Yes: **kazuhide seo

Reviewer #2: No

Reviewer #3: No

---

## [Author Response · Author response to Decision Letter 2]

16 Sep 2025

The reviewer’s comments and suggestions have been carefully addressed in the revised version of the paper. Changes in the revised version have been colored red. We are grateful once again to the associate editor and reviewers for their time and helpful comments, which have played a vital role in improving the quality and presentation of the original manuscript.

---

## [Editor Report · Decision Letter 2]

18 Sep 2025

Cross-modal fusion of brain imaging and clinical data for Parkinson's disease progression prediction

PONE-D-25-29378R2

Dear Dr. Wen,

We’re pleased to inform you that your manuscript has been judged scientifically suitable for publication and will be formally accepted for publication once it meets all outstanding technical requirements.

All concerns have been successfully addressed, and the manuscript has been significantly improved.

Kind regards,

Nima Broomand Lomer, M.D.

Academic Editor

PLOS ONE
---

## [Editor Report · Acceptance letter]

PONE-D-25-29378R2

PLOS ONE

Dear Dr. Wen,

I'm pleased to inform you that your manuscript has been deemed suitable for publication in PLOS ONE. Congratulations! Your manuscript is now being handed over to our production team.

Kind regards,

on behalf of

Dr. Nima Broomand Lomer

Academic Editor

PLOS ONE